# Workplace Bullying Seen from the Perspective of Bystanders: Effects on Engagement and Burnout, Mediating Role of Positive and Negative Affects

**DOI:** 10.3390/ijerph20196821

**Published:** 2023-09-25

**Authors:** Joséphine Pascale Ada Manga, Pascale Desrumaux, Willys N’dong Nguema

**Affiliations:** 1Faculty of Education, University of Ngaoundere, Ngaoundere P.O. Box 454, Cameroon; pascale_ada@yahoo.fr; 2Psychology Department ULR 4072–PSITEC—Psychologie: Interactions, Temps, Emotions, Cognitions, Faculty PsySEF, University Lille, F-59000 Lille, France; willisndong@gmail.com

**Keywords:** workplace bullying, bystanders, psychological health, work engagement, burnout, positive and negative affects

## Abstract

The first and original aim of this study was to measure the impact of workplace bullying (WB) seen from the perspective of the witnesses (bystanders) on the work engagement and the burnout of these bystanders. The second aim was to test the mediating roles of affects between WB seen from the perspective of bystanders and two resulting variables, bystanders’ work engagement and bystanders’ burnout. This study was conducted using self-administered questionnaires with WB bystanders (*n* = 222) from the Cameroonian health sector. The results indicated that positive and negative affects played mediating roles between WB as seen by witnesses and the two resulting variables, burnout and work engagement. This study offers new avenues for intervention on the issue of WB bystanders. In addition to prevention for victims, witnesses experiencing bullying as bystanders in Cameroon must be supported and accompanied by occupational health services, occupational and psychosocial risk prevention workers, psychologists, as well as human resources.

## 1. Introduction

The topic of workplace bullying (WB) is at the heart of several debates [1,2,3,4] and affects a wide range of jobs [1,2,5]. Health care and nursing sectors are particularly affected by bullying acts [6]. According to [7], 20% of these staff report experiencing harassing behaviors regularly and persistently. Indeed, numerous surveys [1] show that social workers, carers, and care providers involved in relational or clinical work are particularly at risk.

Bullying or mobbing is a major social stress [3,4,8,9] with a destructive impact on victims, such as psychological distress [8] and burnout [10,11]. WB leads to disengagement at work [12] and increased absenteeism and intent to quit [7,8].

This complex organizational multi-dimensional phenomenon has often been perceived as simply involving two actors: the victim and the harasser [13]. However, it should be treated as a social relational problem that involves more than two actors [14]. Indeed, [15] point out that WB is not simply a relational problem between two employees but an issue that affects the entire workplace. Indeed, witnesses to such acts are also part of the process and need to be integrated in studies and prevention. In this way, researchers consider those who witness bullying as key because they have more social resources than victims [16], such as getting help from other adults like colleagues, family members, friends, etc.

Paradoxically, little is known about the effects of WB on witnesses and why they are affected [9]. Some rare explanatory studies are qualitative [17] or experimental [18] and some work has attempted to understand the reasons for acting or not acting [19]. Some authors have found that witnessing bullying undermines psychological health (work-related depression and anxiety) and increases insomnia, headaches, and fatigue [20,21].

Understanding the impact of these acts on these witnesses, therefore, becomes crucial.

This study, therefore, focuses on the psychological health of WB witnesses, including burnout and their engagement in work in the health setting. This study has several novel aspects. First, it addresses the issue of WB witnesses in the professional context, using quantitative tools, an aspect that has been explored very little. Indeed, very few studies have measured the psychological health at work of witnesses of bullying, and data on bystanders to harassment are from studies conducted in schools and high schools [22]. Moreover, this study measures not only the bullying seen by the witnesses, but also its impact on the psychological health of these witnesses.

The objective of this study is to approach WB not from the point of view of the victims and/or harassers, but from the point of view of the witnesses, who are present during such negative acts. Although regularly forgotten, this third indirectly involved actor can nevertheless influence this situation and is also influenced, depending on the passive or active role they decide to play [23]. A significant number of people are indirectly exposed to WB [9]. Indeed, 46.5% of workers in the United Kingdom mentioned witnessing WB in the past five years [24], and over 80% of workers in US workplaces say they have witnessed bullying sometimes during their work histories [25]. Another study make a similar finding [26]: 41% report witnessing WB. Figure 1 resume the model of this study.

### 1.1. Psychological Health at Work: Work Engagement and Burnout

Psychological health at work encompasses both a positive and a negative dimension, such as psychological well-being and psychological distress [27,28]. Some studies on psychological health consider burnout and work engagement as negative and positive sides [29,30,31]. Burnout can be considered as a state involving feelings of physical, emotional, and cognitive exhaustion [32]. First, physical exhaustion refers to lower energy to cope with the daily workload. Second, emotional exhaustion refers to the feeling of not being able to invest in relationships at work. Third, cognitive fatigue is characterized by a feeling of cognitive or mental slowness and reduced mental agility [33]. Beyond suffering at work, there is also a certain degree of fulfilment, which will be discussed in terms of engagement to work in the following lines, as a positive side of psychological health at work.

With regard to work engagement, [34] points out that it is among the positive aspects of psychological health that have been most consistently used in the context of work. Authors present work engagement as a positive and fulfilling state of mind towards work [8,35]. It is characterized by three dimensions. Vigor (first dimension) characterizes a high level of energy, a great willingness to invest effort in one’s work, resilience at work, and tenacity in the face of difficult situations. According to [36], this dimension of engagement facilitates the transition from reflection to action, which translates into better performance among committed employees. This means that, with vigor, employees quickly understand what is expected of them and put it into practice. Dedication (second dimension) is characterized by enthusiasm, awareness of the meaning of work, and pride in one’s work. A dedicated person does everything possible to ensure that his or her work goes well. Absorption (third dimension) refers to the attitude of total and deep concentration of the employee at work without any real awareness of the time that is passing [35].

### 1.2. Witnesses of WB and Their Psychological Health in the Health Sector

Ref. [4] characterized WB by the repetition (at least once a week) and duration (at least six months) of exposure to negative behaviors in the conceptualization of WB. Ref. ([1], p. 25) defines it as “a psychological process induced in a work context characterized by a lasting and repeated synergy of destructive acts that undermine the relationships, working conditions and integrity of an employee and result in suffering that can jeopardize his or her psychological and physical health. According to [37], WB has a physical and mental impact on employees and can lead to extreme feelings of distress or hopelessness. WB has significant consequences, both for the victim and for the entire organization, including deterioration in the mental and physical health of targets [1,10], distress [10] , and risk of suicide [12]. Studies have found that WB can lead to burnout [38]. It also impacts satisfaction, engagement, and intent to quit [39]. In addition, it leads to lower work engagement among victims [8]. According to [40], WB is related to depression, anxiety, and burnout. These results are consistent with other studies, showing associations between bullying and burnout [11,41]. 

Although the effects on the target of WB are well documented in the literature, the bystanders or witnesses of WB have not received as much attention [42]. Witnesses of WB, the third actor in WB scenes are, like the victims, also negatively affected by these negative acts at work [43], experience more health problems, and have more unfavorable work attitudes compared to those not involved in bullying [24]. Workplace bullying can decrease work motivation [7,44] and increase the absenteeism, and turnover intention of bullying witnesses [45,46,47] showed that individuals who witnessed bullying tended to report higher substance use, depression, anxiety, and feelings of inferiority. Another study has found that individuals who are involved in multiple roles, including bystanders, are at elevated risk for serious mental health concerns including suicidal ideation ([48]. Bullying affects the well-being of bystanders, leading to psychological, physical, and emotional strain and suffering [10,49,50,51]. 

### 1.3. WB as Seen by Witnesses, Affects, and Psychological Health

Emotions are the organism’s response to a request for adaptation [52,53]. They are keys to understanding employees’ reactions to work. Adaptation arouses positive and negative emotions in individuals with the mission of assessing the situation and preparing appropriate responses or strategies [54]. Positive affect refers to feelings of arousal and pleasantness. Negative affect refers to anger, fear, and guilt. Positive affect plays a crucial role in coping efforts and contributes to physical and psychological well-being. Some authors have suggested the existence of links between WB and emotions at work [18,55,56,57] referred to the existence of an “emotion cocktail”. However, the scientific research conducted on the topic is very limited, as is the range of emotions analyzed. Bullied individuals often experience negative emotions such as fear, shame, anxiety, guilt, doubt, sadness, and humiliation [1,58]. Witnesses of bullying experience more contrasting emotions ranging from no emotion, to empathetic emotions, to even negative emotions such as fear of associating with the person (or group) being bullied, for fear of being bullied back and losing certain benefits or gains [55]. This research [18,55,56] has shown that WB generates negative affect and decreases positive affect. 

Positive affect would promote creativity, cognitive flexibility, work productivity, job satisfaction, as well as optimize work quality [59]. In contrast, employees who experience negative affect at work will tend to show low emotional engagement to work and resign from the company. In their study, [60] found that positive emotional reactions predicted affective engagement and altruistic behavior. Furthermore, [56] found that positive and negative affect play a central (mediating) role in the relationship between WB and its consequences, including job satisfaction, organizational engagement, and intention to leave the organization. 

## 2. Method

### 2.1. Hypothesis of This Study

The following six hypotheses were formulated:

**Hypothesis** **1.**
*Workplace bullying as seen by witnesses is negatively related to work engagement;*


**Hypothesis** **2.**
*Workplace bullying as seen by witnesses is positively related to burnout;*


**Hypothesis** **3.**
*Workplace bullying as seen by witnesses is positively related to negative affect;*


**Hypothesis** **4.**
*Workplace bullying as seen by witnesses is negatively related to positive affect;*


**Hypothesis** **5.**
*Positive and negative affects play mediating roles between workplace bullying as seen by witnesses and work engagement;*


**Hypothesis** **6.**
*Positive and negative affects play mediating roles between workplace bullying as seen by witnesses and burnout.*


### 2.2. Procedure

The participants were called in to answer the questionnaire directly in the hospitals, using five, six, or seven-point Likert scales ranging from totally disagree to totally agree, or highly insufficient to far too many/much, or almost never to almost always. We used different instructions and different response scales in order to minimize response biases, as recommended by [61]. 

Data were collected from seven Cameroonian public and private hospitals in the city of Yaoundé. We have used an anonymous paper form of questionnaires and have invited participants to complete it voluntarily. The purpose of the survey was explained when distributing the questionnaires. Others were given to the heads of the services for those who were absent.

For this data collection, medical and paramedical employees received a letter explaining the purpose of the study (i.e., to expose psychosocial risks in the medical and paramedical profession) and inviting them to complete a paper version of the questionnaire. The letter explained that the participants had to be careful not to give their name or other individual information in their responses, to keep their confidentiality and anonymity, and that their participation was voluntary. The questionnaires were placed directly by the respondent in a sealed envelope and the envelopes were collected directly by the interviewer at the workplace.

### 2.3. Measures 

All measures were administered in French. Properties (means, standard deviations, and latent correlations) are presented in Table 1.

*Work engagement.* The Utrecht Work Engagement Scale-Short form (UWES-9) by [31] was used. This short form consisted of 9 items that measured the three dimensions of work engagement. Each of these dimensions was measured by three items: vigor (e.g., “I am bursting with energy for my work”; α = 0.71), dedication (e.g., “I am passionate about my work”; α = 0.65), and absorption (e.g., “I am literally immersed in my work”; α = 0.73). Participants responded using a frequency scale that ranged from 0 (*never*) to 6 (*always*). Consistent with previous studies ([62], the three dimensions were analyzed individually. A mean score per dimension was calculated. 

*Burnout.* The French version of the Shirom-Melamed Burnout Measure ([63] was composed of 14 items divided into three dimensions, which are physical fatigue (6 items, e.g., “My batteries are dead”, α = 0.79), cognitive weariness (5 items; e.g., I feel I am not thinking clearly”; α = 0.84), and emotional exhaustion (3 items, e.g., “I feel unable to sense the needs of my colleagues and/or patients”; α = 0.74). Each question was associated with a 7-point Likert-type scale ranging from 1 (*never*) to 7 (*always*). In accordance with previous studies, the three dimensions were analyzed separately.

*Workplace Bullying as seen by witnesses*. It was measured with the French version of the Negative Acts Questionnaire (Revised NAQ-R; [64] after we reworded the items so that they would be relevant to the witnesses. An exploratory factor analysis was used to validate the scale. The first version of the scale included 22 items associated with a 5-point Likert-type scale going from 1 (*never*) to 5 (*daily*). After exploratory factor analysis, we retained only 13 items, reflecting work-related WB (4 items; α = 0.72; e.g., “A co-worker has been ordered to do work below his or her level of competence”); person-related WB (5 items; α = 0.81; e.g., “A co-worker has been ignored or facing a hostile reaction when his or her approach”); and bullying by intimidation (4 items; α = 0.73; e.g., “Intimidating behaviors on a co-worker”).

*Positive and negative affect.* Positive Affectivity and Negative Affectivity Schedule (PANAS) [65] consisted of 20 items: 10 positive affect items (e.g., “excited”) and 10 negative affect items (e.g., “pained”) associated with a 5-point Likert-type scale ranging from 1 (*never*) to 5 (*every day*). The alphas were 0.77 for positive affect and 0.75 for negative affect.

To test our hypothesis, we will first present correlations between variables of this study, and after, mediation analyses will be shown.

### 2.4. Participants

A total of 450 questionnaires were distributed to doctors, nurses, nurses’ assistants, and midwives in different hospital workplaces in the town of Yaoundé (Cameroon). Midwives in Cameroon belong to another category of health professionals whose mission is centered on childbirth. This status is intermediate between nurses, doctors, etc. Of the 450 questionnaires, 222 were completely filled, i.e., 49% participation. Our sample was made up exclusively of employees who had witnessed bullying, including 140 women (63.1%) and 82 men (36.9%). The mean age was 35 years (*SD* = 9.1) and their mean job tenure was 4.5 (*SD* = 5). In the sample, 66% were in the public sector, 15% in the private sector, and 19% in the para-public sector. In the sample, 7% were doctors, 46% were nurses, 11% were midwives, 20% were nursing assistants, and 16% were trainees.

The regional ethics committee of the center in the city of Yaoundé gave their approval for this research in the letter CE N° 0125/CRERSHC/2018, signed by Mr. BEYE Casimir.

## 3. Results

In a preliminary analysis part, we examined the descriptive data (means, standard deviations, correlations between variables of this study), and in the Mediation Analyses part (model-testing), we tested hypotheses related to mediation analyses with Hayes and Preacher’s [66] macro model 4 for SPSS. This method tests direct and indirect effects using regressions and a nonparametric bootstrapping procedure generating 10,000 alternative samples and a 95% confidence interval. Mediations were performed using affect as the mediating variable, WB as the VI, and burnout as the DV. In an effort to present smaller tables, we will illustrate the results of the significant relationships.

### 3.1. Descriptive Analysis

According to Table 1, work-related bullying was positively correlated with physical fatigue (*r* = 0.18, *p* < 0.01) and cognitive fatigue (*r* = 0.20, *p* < 0.01). Person-related bullying is positively correlated with physical fatigue (*r* = 0.20, *p* < 0.01), cognitive fatigue (*r* = 0.23, *p* < 0.01), and emotional exhaustion (*r* = 0.24, *p* < 0.001). Bullying by intimidation was positively correlated with cognitive fatigue (*r* = 0.17, *p* < 0.05) and emotional exhaustion (*r* = 0.17, *p* < 0.05). 

Work-related WB correlated negatively with dedication *(r* = −0.18, *p* < 0.01). Person-related bullying also correlates negatively with dedication (*r* = −0.14, *p* < 0.05).

Positive affect is negatively correlated with physical fatigue (*r* = −0.13, *p* < 0.05) and emotional exhaustion (*r* = −0.17, *p* < 0.05), whereas negative affect is positively correlated with fatigue (*r* = 0.28, *p* <.001), weariness (*r* = 0.30, *p* < 0.001), and exhaustion (*r* = 0.35, *p* < 0.001). Positive affect correlated positively with vigor, dedication (*r* = 0.51, *p* < *0*.001) and absorption (*r* = 0.43, *p* < *0*.001). Negative affect correlated negatively with vigor (*r* = −0.25, *p* < 0.001), dedication (*r* = −0.23, *p* < 0.01), and absorption (*r* = −0.19, *p* < 0.01). Work-related bullying correlated positively with negative affect (*r* = 0.22, *p* < 0.01). Person-related bullying is positively correlated with negative affect (*r* = 0.24, *p* <0.001). Bullying by intimidation is negatively correlated with positive affect (*r* = −0.15, *p* < 0.05). 

With regard to the two dependent variables (burnout and commitment) and each of their sub-dimensions, the results of Student’s T indicated that there were no significant differences between men and women. In terms of health professions, the Anova confirmed that there were no differences between health professions (doctors, specialist doctors, midwives, nurses, trainees, care assistants) for the two dependent variables (burnout and commitment) and each of their sub-dimensions. As a result, we have not sub-sampled for gender or health professions.

### 3.2. Mediation Analyses

Mediation results (Table 2) between WB as seen by witnesses and burnout showed that the A link between IV (work-related WB) and MV (negative affect) is significant (β = 0.18 **). The B link between MV (negative affect) and DV (physical fatigue) is significant (β = 0.33 ***). The direct link (C’ link) between IV and DV is not significant (β = 0.13). The total link (C’ link) between IV and DV is significant (β = 0.22 **). The indirect link is significant, β = 0.06, 95% CI [0.01, 0.13]: negative affect has a mediating role between work-related WB and physical fatigue.

Next, the link between IV (work-related WB) and MV (negative affect) (link A) is significant (β = 0.18 **). The link between MV (negative affect) and DV (cognitive fatigue) (link B) is significant (β = 0.38 **). The direct link (link C’) between IV and DV is significant (β = 0.18 *). The total link (C’ link) between IV and DV is significant (β = 0.28 **). The indirect link is significant, β = 0.07, 95% CI [0.02, 0.14]: negative affect plays a mediating role between work-related WB and cognitive fatigue.

In addition, the link between IV (work-related WB) and MV (negative affect) (Link A) is significant (β = 0.18 **). The link between MV (negative affect) and DV (cognitive weariness) (link B) is significant (β = 0.54 ***). The direct link (link C’) between IV and DV is not significant (β = 0.19). The total link (C’ link) between IV and DV is not significant (β = 0.00). The indirect link is significant, β = 0.09, 95% CI [0.03, 0.18]: negative affect plays a mediating role between work-related WB and emotional exhaustion.

The link between IV (person-related WB) and MV (negative affect) (link A) is significant (β = 0.20 ***). The link between MV (negative affect) and DV (physical fatigue) (link B) is significant (β = 0.31 *). The direct link (link C’) between IV and DV is significant (β = 0.17 *). The total link (C’ link) between IV and DV is significant (β = 0.25 **). The indirect link is significant β = 0.06, 95% CI [0.01, 0.13]: negative affect plays a mediating role between person-related WB and physical fatigue.

The link between IV (person-related WB) and MV (negative affect) (link A) is significant (β = 0.20 ***). The link between MV (negative affect) and DV (cognitive weariness) (link B) is significant (β = 0.35 **). The direct link (link C’) between IV and DV is significant (β = 0.24 *). The total link (C’ link) between IV and DV is significant (β = 0.32 ***). The indirect link is significant, β = 0.07, 95% CI [0.01, 0.14]: negative affect plays a mediating role between person-related WB and cognitive fatigue.

The link between IV (person-related WB) and MV (negative affect) (link A) is significant (β = 0.20 ***). The link between MV (negative affect) and DV (emotional exhaustion) (link B) is significant (β = 0.47 ***). The direct link (link C’) between IV and DV is significant (β = 0.20 *). The total link (C’ link) between IV and DV is significant (β = 0.36 ***). The indirect link is significant, (β = 0.09), 95% CI [0.03, 0.18]: negative affect plays a mediating role between person-related WB and emotional exhaustion.

Mediation results (Table 3) between WB as seen by the witnesses and work engagement showed that the link between IV (WB related to work behaviors) and MV (positive affect) (link A) is significant (β = 0.13 *). The link between MV (positive affect) and DV (vigor) (link B) is significant (β = 0.79 ***). The direct link (link C’) between IV and VD is not significant (β = 0.00). The total link (C’ link) between IV and DV is not significant (β = 0.10). The indirect link is significant, β = 0.10., 95% CI [0.01, 0.21]: positive affect plays a mediating role between work behavior-related WB and vigor.

The link between IV (WB related to work behaviors) and MV (positive affect) (link A) is significant (β = 0.13 *). The link between MV (positive affect) and DV (vigor) (link B) is significant (β = 0.72 ***). The direct link (link C’) between IV and VD is not significant (β = 0.01). The total link (C’ link) between IV and DV is not significant (β = 0.07). The indirect link is significant, β = 0.09, 95% CI [0.01, 0.19]: positive affect plays a mediating role between WB related to work behaviors and dedication. 

The link between IV (WB related to work behaviors) and MV (positive affect) (link A) is significant (β = 0.13 *). The link between MV (positive affect) and DV (vigor) (link B) is significant (β = 0.61 ***). The direct link (link C’) between IV and DV is not significant (β = 0.01). The total link (C’ link) between IV and DV is not significant (β = 0.10). The indirect link is significant, β = 0.08, 95% CI [0.01, 0.16]: positive affect plays a mediating role between WB related to work behaviors and absorption.

## 4. Discussion

The purpose of this study was to find out whether the WB experienced indirectly by the witnesses had any effect on their burnout and work engagement, on the one hand, and on the other hand, to see if affects would mediate this relationship. The correlational analysis allowed us to highlight the links between the variables in the study. There is indeed a positive link between WB and work engagement, particularly in the relationship between WB related to work, WB related to the person, and the work engagement [62,67,68,69]. 

Hypothesis H1 of the link between WB and work engagement is partially validated for the dedication. The H2 hypothesis stipulating a link between WB and burnout is validated. This result is in line with another study which revealed that witnesses of WB can also be affected by the actions between the victim and the harasser [23]. This result is found in the study by [70]. Indeed, negative affect and burnout increase simultaneously. Positive affect is related to an increase in work engagement. This result is consistent with previous studies that emphasize that negative affect has a negative impact on workers while positive affect is beneficial for workers. Specifically, positive affect increases job productivity, job satisfaction, and job quality [59,71]. On the other hand, negative affect has harmful effects, such as job dissatisfaction, low emotional engagement to work, and intentions to leave the company, etc. [59]. 

This study highlights the link between WB as seen by witnesses and the negative and positive affects of witnesses. WB through intimidation lowers witnesses’ positive affect. The results reveal WB promotes the occurrence of negative affect and the decrease of positive affect. This study provides additional information complements, for witnesses, in the existing research showing negative consequences of WB in employees such as signs of psychological distress and burnout [10,11]. Our study also completes studies in the healthcare sector showing that witnessing bullying has negative consequences on quality of care, on work engagement, and on turnover intentions [45,46].

Negative affect would be related to all WB modalities which validate hypothesis H4. As for hypothesis H3, it would be partially validated insofar as the only link that exists is between WB by intimidation and positive affect. Both negative and positive affect are correlated with all dimensions of work engagement and burnout. It appears from the mediation analyses that negative affect plays mediating roles between WB (work-related and personal) and the three dimensions of burnout. Positive affect plays mediating roles between WB by intimidation and the three dimensions of work engagement. Hypotheses H5 and H6 are, therefore, partially validated. This study confirms the mediating role of affect in the relationships between WB and burnout, and work engagement. Indeed, negative affect played a mediating role in the relationship between WB (work- and person-related WB) and burnout, whereas positive affect played a mediating role in the relationship between WB (bullying WB) and work engagement. 

This study of WB bystanders helps confirm that bystanders also suffer from bullying, as do frontline actors, including victims. Indeed, witnessing WB is associated with burnout in these witnesses. These findings offer a complementary view to previous studies linking WB to burnout among victims [10,11] and to work engagement [62]. They shed light on the understanding of the links between WB and burnout: WB promotes the occurrence of negative affect, which in turn depletes the emotional, physical, and cognitive resources of witness employees. The saving role of positive affect is also to be taken into consideration because it promotes the work engagement of WB witnesses.

### Limitations and Perspectives

The use of a self-reported questionnaire generates biases (e.g., social desirability, halo...). The use of self-reported responses increases the risk of collinearity and may increase the common variance [72]. The cross-sectional design does not allow for causality to be established between variables. To establish causality requires repeated measures or a longitudinal study. To increase the external validity of the study, a sample that includes other occupational areas would be recommended. A qualitative approach could highlight new aspects that would not have been identified by a quantitative approach.

This research has the advantage of focusing on bystanders, who may have the opportunity to intervene in an attempt to stop the abuse. Increasing the number of studies of the many witnesses will open up new avenues of intervention. As bullying becomes protean, research that considers cyber bullying [10] would be recommended to explore the role of bystander behaviors at the digital level and measure the consequences. Finally, as [73] showed that workplace insecurity mediated the role of bullying on engagement and health, it would be interesting to explore this mediating role for the witnesses.

## 5. Practical Implications and Conclusions

This study offers avenues for intervention on the issue of WB. Indeed, this study reinforces the idea that a WB scene involves more than two actors, other than the victims and the harassers. Witnesses are inseparable actors in this scene. They may be vicarious victims or passive victims [74] because WB’s acts impact their well-being and psychological health [75]. Thus, this study demonstrates the need to include WB seen by witnesses in health and safety prevention policies within organizations. 

Indeed, the implementation of prevention measures on the WB must integrate these numerous actors who can change the outcome of these acts. They can either put an end to it or encourage its continuation. Thus, the witness must be consulted every time managers want to implement concrete actions to counteract WB within the organizations. Involving them in this process would lead to results that would really curb this phenomenon.

Moreover, these third actors in the WB scene suffer from WB as victims, although they are indirectly involved. Experiencing WB as a witness decreases their engagement to their work, especially in terms of dedication according to our results, and increases their level of burnout. Cleaning up the work environment will have to be a priority for all employees because WB greatly harms the performance of the entire organization [70]. This remediation work involves consultation, enforcement, reframing, and implementation of sanctions by HRDs and top management. 

In addition, positive and negative affects must also be integrated when dealing with the issue of WB. Previous research has presented the merits of positive affect in the workplace and the harmful effects of negative affect in the daily lives of workers. Thus, preventing and implementing devices to combat WB requires the integration of emotional aspects present in the work environment for an optimal result in terms of preserving workers’ health. 

Ultimately, negative affect plays a pivotal role in this deterioration of their psychological health, and positive affect is lifesaving for their psychological health. Therefore, preventive measures should be taken to prevent risks to the psychological health of witnesses of WB, as well as victims. In addition to measures to protect victims, witnesses must be accompanied by human resources services, occupational health services, risk prevention services, and psychologists [1]. This support will help preserve and protect their psychological health. Witnesses can then participate in the protection and well-being of WB victims.

## Figures and Tables

**Figure 1 ijerph-20-06821-f001:**
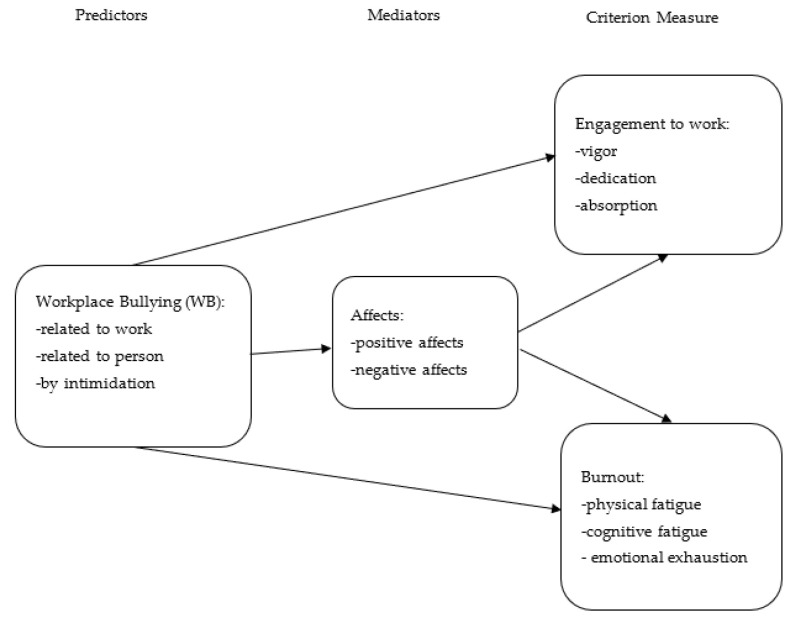
Hypothesized Model: Effect of WB on psychological health (engagement to work and burnout) via affects.

**Table 1 ijerph-20-06821-t001:** Means, standard deviations, and correlation between variables.

	*M*	*SD*	1	2	3	4	5	6	7	8	9	10	11
1. Eng-vig	4.59/6	1.08	**0.71**										
2. Eng-dev	4.88/6	0.99	0.72 ***	**0.65**									
3. Eng-abs	4.66/6	0.97	0.66 ***	0.62 ***	**0.73**								
4. Bo-PF	3.11/7	0.95	−0.25 ***	−0.20 ***	−0.09	**0.79**							
5. Bo-CF	2.92/7	1.07	−0.24 ***	−0.20 **	−0.15 *	0.56 ***	**0.84**						
6. Bo-EE	2.53/7	1.15	−0.20 **	−0.23 **	−0.20 ***	0.43 ***	0.47 ***	**0.74**					
7. WB-work	2.43/5	0.76	−0.06	−0.18 **	0.00	0.18 **	0.20 **	0.13	**0.72**				
8. WB-person	2.50/5	0.77	−0.03	−0.14 *	−0.03	0.20 **	0.23 **	0.24 ***	0.58 ***	**0.81**			
9. WB-int	2.73/5	0.75	−0.07	0.06	0.08	0.10	0.17 *	0.17 *	0.45 ***	0.55 ***	**0.73**		
10. Positive affect	3.98/5	0.65	0.51 ***	0.51 ***	0.43 ***	−0.13 *	−0.13	−0.17 *	−0.02	0.05	0.15 *	**0.77**	
11. Negative affect	2.36/5	0.63	−0.25 ***	−0.23 **	−0.19 **	0.28 ***	0.30 ***	0.35 ***	0.22 **	0.24 ***	0.08	0.39 ***	**0.75**

*n* = 222; *M*: Mean; *SD*: Standard Deviation; Eng-Vi: Engagement to Work-Vigor; Eng-De: Engagement to Work-Dedication; EngAb: Engagement to Work-Absorption; Bo-PF: Burnout-Physical Fatigue; Bo-CF: Burnout-Cognitive Fatigue; Bo-EE: Burnout-Emotional Exhaustion; WB-work: Workplace Bullying related to work; WB-person: Workplace Bullying related to person; WB-Int: Workplace Bullying by intimidation; Bolded Cronbach’s Alpha on the diagonal; * *p* < 0.05, ** *p* < 0.01, *** *p* < 0.001.

**Table 2 ijerph-20-06821-t002:** Mediations of affects between WB as seen by witnesses and burnout.

IV	Med V	Effect of IV on MV (Link A)	Effect of MV on DV (Link B)	Total Effect (Link C)	Direct Effect (Link C’)	Indirect Effect	CI LL UL for Indirect Effect
Burnout-Physical Fatigue (DV)							
WB-wor							
	Pos. affect	−0.02	−0.05	0.22 **	0.13	0.00	[−0.01, 0.01]
	Neg. affect	0.18 **	0.33 ***	0.22 **	0.13	0.06 ^a^	[0.01, 0.13]
WB–per							
	Pos. affect	0.04	−0.07	0.25 **	0.17 *	−0.00	[−0.03, 0.01]
	Neg. affect	0.20 ***	0.31 **	0.25 **	0.17 *	0.06 ^a^	[0.01, 0.13]
WB–Int							
	Pos. affect	0.13 *	−0.05	0.13	0.06	−0.01	[−0.04, 0.02]
	Neg. affect	0.07	0.36	0.13	0.06	0.03	[−0.02, 0.08]
Burnout-Cognitive Fatigue (DV)							
WB-wor							
	Pos. affect	−0.02	−0.04	0.28 **	0.18 *	0.00	[−0.02, 0.02]
	Neg. affect	0.18 **	0.38 **	0.28 **	0.18 *	0.07 ^a^	[0.02, 0.14]
WB-per							
	Pos. affect	0.04	−0.07	0.32 ***	0.24 *	−0.00	[−0.03, 0.01]
	Neg. affect	0.20 ***	0.35 **	0.32 ***	0.24 *	0.07 ^a^	[0.01, 0.14]
WB-int							
	Pos. affect	0.13 *	−0.06	0.25 *	0.18	−0.01	[−0.04, 0.03]
	Neg. affect	0.07	0.41 **	0.25 *	0.18	0.03	[−0.02, 0.09]
Burnout-Emotional Exhaustion (DV)							
WB-wor							
	Pos. affect	−0.02	−0.03 **	0.19	0.00	0.00	[−0.02, 0.02]
	Neg. affect	0.18 **	0.54 ***	0.19	0.00	0.09 ^a^	[0.03, 0.18]
WB-per							
	Pos. affect	0.04	−0.08	0.36 ***	0.20 *	−0.00	[−0.03, 0.01]
	Neg. affect	0.20 ***	0.47 ***	0.36 ***	0.20 *	0.09 ^a^	[0.03, 0.18]
WB-Int							
	Pos. affect	0.13 *	−0.07	0.26 *	0.15	−0.01	[−0.05, 0.03]
	Neg. affect	0.07	0.52 ***	0.26 *	0.15	0.04	[−0.02, 0.11]

WB: workplace bullying; WB-wor.: work-related WB, WB-per.: person-related WB; WB-int.: WB by intimidation; pos. affect: positive affect; neg. affect: negative affect; CI: confidence interval; LL: lower limit, UL: upper limit; Bootstrapping *n* = 10,000. ^a^ *p* < 0.05 (bootstrapping 95% CI does not include zero). * *p* < 0.05, ** *p* < 0.01, *** *p* < 0.001.

**Table 3 ijerph-20-06821-t003:** Mediations of affects between witnesses’ experienced WB and work engagement.

IV	Med V	Effect of IV on MV (Link A)	Effect of MV on DV (Link B)	Total Effect (Link C)	Direct Effect (Link C’)	Indirect Effect	CI LL UL for Indirect Effect
Engagement to work-Vigor (DV)							
WB-wor							
	Pos. affect	−0.02	0.80 ***	−0.08	−0.05	−0.01	[−0.12, 0.07]
	Neg. affect	0.18 **	−0.09	−0.08	−0.05	−0.02	[−0.06, 0.03]
WB-per							
	Pos. affect	0.05	0.51 ***	−0.03	−0.06	0.03	[−0.03, 0.10]
	Neg. affect	0.24 ***	0.25 ***	−0.03	−0.06	0.01	[−0.19, 0.28]
WB-int							
	Pos. affect	0.13 *	0.79 ***	0.10	−0.00	0.10 ^a^	[0.01, 0.21]
	Neg. affect	0.07	−0.01	0.10	−0.00	−0.01	[−0.04, 0.01]
Engagement to work–Dedication (DV)							
WB-wor							
	Pos. affect	−0.02	0.74 ***	−0.23 **	−0.19	−0.01	[−0.11, 0.07]
	Neg. affect	0.18 **	0.02	−0.23 **	−0.19	0.00	[−0.04, 0.05]
WB-per							
	Pos. affect	0.05	0.52 ***	−0.12	−0.07	−0.00	[−0.04, 0.05]
	Neg. affect	0.24 ***	−0.21 **	−0.12	−0.07	−0.06	[−0.03, 0.10]
WB-int							
	Pos. affect	0.13 *	0.72 ***	0.07	0.01	0.09 ^a^	[0.01, 0.19]
	Neg. affect	0.07	−0.03	0.07	0.01	−0.00	[−0.03, 0.02]
Engagement to Work-Absorption (DV)							
WB-wor							
	Pos. affect	−0.02	0.62 ***	−0.00	0.01	−0.01	[−0.09, 0.06]
	Neg. affect	0.18 **	0.54 ***	−0.00	0.01	−0.01	[−0.06, 0.03]
WB-per							
	Pos. affect	0.05	0.43 ***	0.00	−0.19	0.02	[−0.22, 0.16]
	Neg. affect	0.24 ***	0.20 **	0.00	0.05	0.05	[−0.14, 0.29]
WB–int							
	Pos. affect	0.13 *	0.61 ***	0.10	0.01	0.08 ^a^	[0.01, 0.16]
	Neg. affect	0.07	−0.07	0.10	0.01	−0.01	[−0.03, 0.01]

WB: workplace bullying; WB-wor.: work-related WB, WB-per.: person-related WB; WB-int.: WB by intimidation; pos. affect: positive affect; neg. affect: negative affect; CI: confidence interval; LL: lower limit, UL: upper limit; Bootstrapping; *n* = 10,000. ^a^
*p* < 0.05 (bootstrapping 95% CI does not include zero). * *p* < 0.05, ** *p* < 0.01, *** *p* < 0.001.

## Data Availability

Data are available on request from pascale_ada@yahoo.fr.

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
