# Peer review of "Workplace Bullying Seen from the Perspective of Bystanders: Effects on Engagement and Burnout, Mediating Role of Positive and Negative Affects"

_ijerph, 2023, doi:10.3390/ijerph20196821_

Round 1

Reviewer 1 Report

1. Showcase and compare more relevant references to highlight the innovation and contribution of your research.

2. Fine-tune the presentation of this paper, especially the format according to IJERPH template.

3. Improve the discussion and the conclusion. For example, is there any more content of 4.2 or 4.x after listing 4.1. Limitations and perspectives?

Moderate editing of English language required

Author Response

Dear Reviewer,

The responses are in the attached document.

Kind regards,

The authors

Reviewer 2 Report

Please see my detailed comments on the attached pdf as indicted in blue. 

Thank you for the opportunity to review this article - I find it very valuable and interesting.

Can improve - I indicated a few places that needs attention. 

Author Response

Reviewer 2

Please see my detailed comments on the attached pdf as indicted in blue. 

Thank you for the opportunity to review this article - I find it very valuable and interesting.

Title: Workplace Bullying Seen By Bystanders: Effects on Engagement and Burnout, Mediating role of Positive and Negative Affects

Your suggestion for the title: "WB as seen from the Perspective of...."

Response: Workplace Bullying Seen from the Perspective of Bystanders: Effects on Engagement and Burnout, Mediating role of Positive and Negative Affects

Abstract: Is "the" WB really needed? Only refer to WB as a construct

Response: The first and original aim of this study was to measure the impact of workplace bullying (WB) seen from the perspective of the witnesses (bystanders) on the work engagement and the burnout of these bystanders. The second aim was to test the mediating roles of affects between WB seen from the perspective of bystanders and two resulting variables, bystanders’ work engagement and bystanders’ burnout. 

Keywords: Consider including both positive and negative affect here

Response: Workplace bullying; bystanders; psychological health; work engagement; burnout; positive and negative affects

Introduction: Explain why used in conjunction with "mobbing" - just motivate (line 33)

Response: Mobbing or bullying are interchangeable terms commonly used in the scientific literature to describe hostile and malicious behaviour towards an individual in a social environment. This concept was introduced by Leymann (1993) to refer to the 45 acts of the harasser.

Suggestion: Multi-dimensional phenomenon? (Line 38)

Response: we have modified the sentence as follow: This complex organizational multi-dimensional phenomenon, has often been mistakenly perceived as simply involving two actors

Integrated - past tense (line 44)

Response: In this way, researchers considered them, those who were witness bullying, « as a key » because they had more social resources than victims (Kim, 2014).

Explain what is meant by "social resources" here please - just elaborate shortly... (Line 44)

Response: In this way, researchers consider them, those who witness bullying, « as a key » because they have more social resources than victims (Kim, 2014), such as getting help from another adult, like colleague, family members, friends, etc.

For the UK yes, but what about other countries on a global scale? Maybe just refer to the tendencies on a world-wide scale? (Lines 52-53)

Response: Indeed, 46.5% of workers in the United Kingdom mentioned witnessing WB in the past five years (Hoel & Cooper, 2000), over 80% of workers, in US workplaces, say they have witnessed bullying sometime during their work histories (Lutgen-Sandvik 2006).

Please define personnel here: whom will it include in the context of this study? (line 70)

Response: Moreover, this study measures not only the bullying seen by the witnesses but also its impact on the psychological health of these witnesses.

Should negative psychological health not be referred to as psychological ill-health rather? (Lines 75-77)

Response:  The term negative psychological health at work refers to the negative side often measured in the literature in terms of distress at work and/or burnout (Boudrias et al., 2011, 2014; Desrumaux, 2015). It is in this sense that it is called negative as opposed to positive psychological health referring to well-being at work. Admittedly, the negative side ultimately corresponds to poor psychological health, but the usual term is negative psychological health at work.

Elaborate a bit more here on what is meant by: .."transition from reflection to action..." (lines 94-95)

Response: According to Demerouti and Cropanzano (2010), this dimension of engagement facilitates the transition from reflection to action, which translates into better performance among committed employees. This means that, with vigor, employees quickly understand what is expected of them and put it into practice.

...as much attention...rather than focus - suggestion only. (Line 120)

Response: Although the effects on the target of workplace bullying are well documented in the literature, the bystanders or witnesses of WB have not received as much attention (Glaso et al., 2009).

Hypothesis 2: by...must be added here

Response: Hypothesis 2: Workplace bullying as seen by witnesses is positively related to burnout.

Combine with recent source (lines 135-136)

Response: we have added these references

Hsieh, Y. H., Wang, H. H., & Ma, S. C. (2019). The mediating role of self-efficacy in the relationship between workplace bullying, mental health and an intention to leave among nurses in Taiwan International journal of occupational medicine and environmental health, 32(2), 245-254. https://doi.org/10.13075/ijomeh.1896.01322

Kim, Y., Lee E., & Lee H. (2020). Correction: Association between workplace bullying and burnout, professional quality of life, and turnover intention among clinical nurses. PLOS ONE 15(1): e0228124. https://doi.org/10.1371/journal.pone.0228124

This can also be seen in high attrition rates – high turnover-intention (line 160)

Response:  we have added the following sentence and references:

Workplace bullying can decrease work motivation (Fernet et al., 2015; Trépanier et al., 2013a) and increase absenteeism, turnover intention of bullying witnesses (Holm et al., 2023; Nielsen et al., 2021).

Why exploitable? – please give a short explanation? (Line 228)

Response: At the end, 222 were completely filled.

Suggestion: It might be interesting to maybe focus in another article on only the positive or the negative aspects, separately. (Lines 335-340)

Response: Yes, it will be very interesting to focus on only the positive or the negative aspects, separately. However, for this article our aim was to quantify both aspects of affect (positive and negative) in order to determine which of these dimensions offers the best explanation of the relationship between the independent variable and the dependent variables.

Please also include more recent sources here to support your argument and strengthen it more. (Line 352)

Response:

Gadi, P. D., & Kee, D. M. H. (2020). Workplace bullying, human resource management practices, and turnover intention: the mediating effect of work engagement: evidence of Nigeria. American Journal of Business, 36(1), 62-83. https://doi-org/10.1108/AJB-08-2020-0135 

Hameed, F., Ambreen, G. and Awan, Y. (2023). "Relationship between workplace bullying and work engagement: education sector of Pakistan", Evidence-based HRM, Vol. ahead-of-print No. ahead-of-print. https://doi-org.ressources-electroniques.univ-lille.fr/10.1108/EBHRM-07-2022-0161

Hamel, J. F., Iodice, P., Radic, K., & Scrima, F. (2023). The reverse buffering effect of workplace attachment style on the relationship between workplace bullying and work engagement. Frontiers in psychology, 14, 1112864. https://doi.org/10.3389/fpsyg.2023.1112864

Is this the only highlight of this study? Is the positive side not also a part of your study's purpose? (Line 353)

Response: This study highlights the link between WB as seen by witnesses and the negative and positive affects of witnesses.

Remark: It will be interesting to see what insights a pure qualitative study might reveal (Line 378)

Response: Your observation is excellent. It would be extremely interesting to explore these elements in depth. We added the following sentence in the limitations. A qualitative approach could highlight new aspects that would not have been identified by a quantitative approach.

Reviewer 3 Report

I was invited to revise the paper (ijerph-2496574) entitled Workplace Bullying Seen By Bystanders: Effects on Engagement and Burnout, Mediating role of Positive and Negative Af-3 affect

I have to say that this is a very well written manuscript and it is a nice and original study of this topic.

However, to improve this article, some comments and suggestions can see below. Firstly, I would like to emphasize some global aspects. Secondly, I will take in account particular aspects in each topic of interest.

Thus:

·        Could the term used "medical sector" be replaced by "health sector"? because it is not just the doctors surveyed but also the nurses and midwives.

·        Also, regarding the concept of "midwife", it would be good to clarify whether it is: a nurse, doctor, nursing assistant or other? In comparison with other countries (in other future studies), this professional category should be more objective.

·        The use of acronyms in scientific writing is very common in the field of health and wellness. However, from the moment it is used, the word(s) that gave rise to it should never be repeated. We can see that the acronym WB is often neglected. It will be better to uniform throughout the text.

ABSTRACT

Last sentence: Witnesses experiencing bullying as bystanders must be supported and accompanied by occupational health services, occupational and psychosocial risk prevention workers, psychologists, as well as human resources. Is this practice common in Cameroon with bulling victims? Or should only these bystanders will be monitored (supported and accompanied by occupational health services)? it is not clear.

INTRODUTION

Verifica-se uma miscelânea de assuntos (introdução e métodos). Os autores deverão reescrever esta secção de modo a escrever os objetivos e as hipóteses do estudo após a escrita do estado da arte e a apresentação das lacunas existentes no conhecimento acerca do problema que querem trabalhar (resolver).

Thus:

Page 2, Lines 47- 55 – Move these objectives to the end of this section.

Page 4, Lines 101- 133 - topic 1.2. Witnesses of Workplace Bullying and Their Psychological Health, it is necessary to clarify what type of Workers the studies refer to. At the end of this topic, the hypotheses of this investigation are described. It would be advisable to put it in the methods section.

Page 4, Lines 154- 170 - the authors describe four more research hypotheses that should be relocated

METHOD

Page 4, Lines 173, 179 – Were the hospitals selected all the ones that exist in the city of Yaoundé? How many? It would be interesting to present the total number of health professionals (doctors, nurses and midwives that exist in these places).

Page 6, Lines 191- 224 – The authors present the measures they will use very well. However, they do not mention the statistical analyzes they are going to do and how. When you get to the results, you get to know. I propose to rewrite this section as well.

Page 6, Line 226 – You shouldn't start a sentence with a number. The same happens on line 230. Please replace them with their description. Nor is it clear why 450 questionnaires were distributed. Clarify, please.

The authors need to add the opinion from an ethics committee, please add their number and name in this section and annex the statement to article.

RESULTS

Page 7, Lines 259 – 266 – Wy is there need to use the bootstrapping method in this study? Clarify, please. Move these statistical procedures to the methods section.

DISCUSSION

The authors mention in the methods that doctors, nurses and midwives, men and women were included. It seems to me that a big gap in this study refers to this very thing. Are these results identical between gender and professions or are they very different? do the authors have these results?

Taking into account the lack of an ethical opinion for the execution of this investigation, the results and the discussion have to be read with great caution.

REFERENCES

Only 16% (11/69) and 35% (24/69) of references were published in the last 5 and 10 years respectively was carried out. Maybe authors can do it an update!? Please, see that possibility.

Attention, please to reference 30.

Author Response

Reviewer 3

I was invited to revise the paper (ijerph-2496574) entitled Workplace Bullying Seen By Bystanders: Effects on Engagement and Burnout, Mediating role of Positive and Negative Af-3 affect

I have to say that this is a very well written manuscript and it is a nice and original study of this topic.

However, to improve this article, some comments and suggestions can see below. Firstly, I would like to emphasize some global aspects. Secondly, I will take in account particular aspects in each topic of interest.

Thus:

  • Could the term used "medical sector" be replaced by "health sector"? because it is not just the doctors surveyed but also the nurses and midwives.

Response: Yes, it is done. We have modified the word in the paper.

  • Also, regarding the concept of "midwife", it would be good to clarify whether it is: a nurse, doctor, nursing assistant or other? In comparison with other countries (in other future studies), this professional category should be more objective.

Response: Midwives in Cameroon belong to another category of health professionals whose mission is centered on childbirth. This status is intermediate between nurses, doctors, etc.

  • The use of acronyms in scientific writing is very common in the field of health and wellness. However, from the moment it is used, the word(s) that gave rise to it should never be repeated. We can see that the acronym WB is often neglected. It will be better to uniform throughout the text.

Response: It is done

ABSTRACT

Last sentence: Witnesses experiencing bullying as bystanders must be supported and accompanied by occupational health services, occupational and psychosocial risk prevention workers, psychologists, as well as human resources. Is this practice common in Cameroon with bulling victims? Or should only these bystanders will be monitored (supported and accompanied by occupational health services)? it is not clear.

Response: We have modified the abstract. The sentence has been clarified

In addition to prevention for victims, witnesses experiencing bullying as bystanders in Cameroon must be supported and accompanied by occupational health services, occupational and psychosocial risk prevention workers, psychologists, as well as human resources.

INTRODUTION

Verifica-se uma miscelânea de assuntos (introdução e métodos). Os autores deverão reescrever esta secção de modo a escrever os objetivos e as hipóteses do estudo após a escrita do estado da arte e a apresentação das lacunas existentes no conhecimento acerca do problema que querem trabalhar (resolver).

Thus: Page 2, Lines 47- 55 – Move these objectives to the end of this section.

Response: The modifications have been made.

 Page 4, Lines 101- 133 - topic 1.2. Witnesses of Workplace Bullying and Their Psychological Health, it is necessary to clarify what type of Workers the studies refer to. At the end of this topic, the hypotheses of this investigation are described. It would be advisable to put it in the methods section.

Response: we have put them over there.

Page 4, Lines 154- 170 - the authors describe four more research hypotheses that should be relocated

Response: we have relocated them.

METHOD

Page 4, Lines 173, 179 – Were the hospitals selected all the ones that exist in the city of Yaoundé? How many? It would be interesting to present the total number of health professionals (doctors, nurses and midwives that exist in these places).

Response: we have collected data from seven hospitals. In the city of Yaoundé, we have more than these one.

Data were collected in the following hospitals:

1-Yaoundé University Hospital Centre

2- Cité verte district hospital

3- Biyem assi district hospital

4- Efoulan district hospital

5- Nkolndongo District Hospital

6- Jourdain Hospital

7- Jamot Hospital

For reasons of confidentiality, we prefer not to display the names of hospitals in the article.

we asked the relevant authorities for the total number of health professionals (doctors, nurses and midwives that exist in these places) but but we were not provided with these job-related data. Finally, at the level of each hospital, we were unable to obtain statistics on the health professions in each hospital.

 Page 6, Lines 191- 224 – The authors present the measures they will use very well. However, they do not mention the statistical analyzes they are going to do and how. When you get to the results, you get to know. I propose to rewrite this section as well.

Response: we added theses sentences at the end of this section:

In a preliminary analysis part, we examined the descriptive data (means, standard deviations, correlations between variables of this study) and in the Mediation Analyses (model-testing part), we tested hypotheses related to mediations analyses with Hayes and Preacher's (2014) macro model 4 for SPSS. This method tests direct and indirect effects using regressions and a nonparametric bootstrapping procedure generating 10,000 alternative samples and a 95% confidence interval. Mediations were performed using affect as the mediating variable, WB as the VI, and burnout as the DV. In an effort to present smaller tables, we will illustrate the results of the significant relationships.

Page 6, Line 226 – You shouldn't start a sentence with a number. The same happens on line 230.

Response: we have corrected it

Nor is it clear why 450 questionnaires were distributed. Clarify, please.

Response: we have corrected it

The number of 450 questionnaires is what we gave effectively: 450 questionnaires were distributed, and of the 450 people who received a questionnaire, 222 were completed in full.

The authors need to add the opinion from an ethics committee, please add their number and name in this section and annex the statement to article.

Response: The Regional ethics comitee of the center in the city of Yaoundé gave their approval for this research in the letter CE N° 0125/CRERSHC/2018, signed by Mr. BEYE Casimir.  

RESULTS

Page 7, Lines 259 – 266 – Wy is there need to use the bootstrapping method in this study? Clarify, please. Move these statistical procedures to the methods section.

 Response:

Explanation of the use of the bootstrapping method: It’s a statistical procedure by Hayes and Preacher's (2014) that allows for the calculation of standard errors, confidence intervals, and hypothesis testing.

“This method tests direct and indirect effects using regressions and a nonparametric bootstrapping procedure generating 10,000 alternative samples and a 95% confidence interval. Mediations were performed using affect as the mediating variable, WB as the VI, and burnout as the DV. In an effort to present smaller tables, we will illustrate the results of the significant relationships.”

We have added these sentences and moved it just after the title “results”.

DISCUSSION

The authors mention in the methods that doctors, nurses and midwives, men and women were included. It seems to me that a big gap in this study refers to this very thing. Are these results identical between gender and professions or are they very different? do the authors have these results?

Response:

Concerning the comparison of men and women with regard to the two dependent variables and each of their sub-dimensions (burnout and commitment), the results of Student's T indicate that there is no significance in the different calculations. In terms of professions, the anovas confirm that there is no difference between health professions (doctors, specialist doctors, midwives, nurses, trainees, care assistants). As a result, we have not sub-sampled for gender or profession.

The results are presented for 222 bystanders from the health sector who are doctors, medical specialists, midwives, nurses, trainees, care assistants.

Taking into account the lack of an ethical opinion for the execution of this investigation, the results and the discussion have to be read with great caution.

 Response: We received the letter of approval for this research by the Regional Ethics Comitee of the Center, in the city of Yaoundé.

REFERENCES

Only 16% (11/69) and 35% (24/69) of references were published in the last 5 and 10 years respectively was carried out. Maybe authors can do it an update!? Please, see that possibility.

Response: we have updated it.

Attention, please to reference 30.

Response: We have modified it.

Round 2

Reviewer 3 Report

No comments. The article is now fine.